# Chest X-ray Score and Frailty as Predictors of In-Hospital Mortality in Older Adults with COVID-19

**DOI:** 10.3390/jcm10132965

**Published:** 2021-07-01

**Authors:** Sara Cecchini, Mirko Di Rosa, Luca Soraci, Alessia Fumagalli, Clementina Misuraca, Daniele Colombo, Iacopo Piomboni, Francesca Carnevali, Enrico Paci, Roberta Galeazzi, Piero Giordano, Massimiliano Fedecostante, Antonio Cherubini, Fabrizia Lattanzio

**Affiliations:** 1Department of Radiology, IRCCS INRCA, 60127 Ancona, Italy; s.cecchini@inrca.it (S.C.); f.carnevali@inrca.it (F.C.); e.paci@inrca.it (E.P.); 2Unit of Geriatric Pharmacoepidemiology and Biostatistics, IRCCS INRCA, 60124 Ancona, Italy; 3Unit of Geriatric Medicine, IRCSS INRCA, 87100 Cosenza, Italy; l.soraci@inrca.it; 4Respiratory Unit, IRCCS INRCA, 23880 Casatenovo, Italy; a.fumagalli@inrca.it (A.F.); c.misuraca@inrca.it (C.M.); d.colombo@inrca.it (D.C.); 5Department of Medicine and Health Sciences, University of Molise, 86100 Campobasso, Italy; i.piomboni@studenti.unimol.it; 6Laboratory of Chemical-Clinical and Molecular Analysis, IRCCS INRCA, 60127 Ancona, Italy; r.galeazzi@inrca.it; 7Internal Medicine and Geriatrics, Hypertension Excellence Centre of the European Society of Hypertension, IRCCS INRCA, 60127 Ancona, Italy; p.giordano@inrca.it; 8Geriatria, Accettazione Geriatrica e Centro di Ricerca per L’invecchiamento, IRCCS INRCA, 60127 Ancona, Italy; m.fedecostante@inrca.it (M.F.); a.cherubini@inrca.it (A.C.); 9Scientific Direction, IRCCS INRCA, 60124 Ancona, Italy; f.lattanzio@inrca.it

**Keywords:** chest radiographic score, COVID-19 pneumonia, frailty, in-hospital mortality

## Abstract

Background. The purpose of this study was to evaluate the prognostic impact of chest X-ray (CXR) score, frailty, and clinical and laboratory data on in-hospital mortality of hospitalized older patients with COVID-19. Methods. This retrospective study included 122 patients 65 years or older with positive reverse transcription polymerase chain reaction for severe acute respiratory syndrome coronavirus-2 (SARS-CoV-2) and with availability to CXRs on admission. The primary outcome of the study was in-hospital mortality. Statistical analysis was conducted using Cox regression. The predictive ability of the CXR score was compared with the Clinical Frailty Scale (CFS) and fever data using Area Under the Curve (AUC) and net reclassification improvement (NRI) statistics. Results. Of 122 patients, 67 died during hospital stay (54.9%). The CXR score (HR: 1.16, 95% CI, 1.04–1.28), CFS (HR: 1.27; 95% CI, 1.09–1.47), and presence of fever (HR: 1.75; 95% CI, 1.03–2.97) were significant predictors of in-hospital mortality. The addition of both the CFS and presence of fever to the CXR score significantly improved the prediction of in-hospital mortality (NRI, 0.460; 95% CI, 0.102 to 0.888; AUC difference: 0.117; 95% CI, 0.041 to 0.192, *p* = 0.003). Conclusions. CXR score, CFS, and presence of fever were the main predictors of in-hospital mortality in our cohort of hospitalized older patients with COVID-19. Adding frailty and presence of fever to the CXR score statistically improved predictive accuracy compared to single risk factors.

## 1. Introduction

The recent outbreak of the novel coronavirus disease in 2019 (COVID-19) has endangered the well-being of healthcare systems worldwide. As of 31 May 2021, the number of cases in Italy has reached more than 4.2 million, with more than 124,000 deaths attributed to COVID-19 [1]. Since the start of the pandemic, older patients exhibited susceptibility to developing more aggressive disease courses and were at a higher risk of mortality related to the disease [2,3]. Several multidimensional scoring systems have been proposed for risk stratification in hospitalized older COVID-19 patients [4,5,6] and were further associated with decreased in-hospital survival and accelerated clinical deterioration [7]; age, respiratory function, laboratory data, and the presence of comorbidities and neurological functions were the main predictors used [4,5,6,8] as they were related to a worse prognosis in this setting [4,5,6,8,9,10].

Another valuable prognostic factor determined to be associated with in-hospital mortality was the radiological severity of lung involvement during COVID-19 pneumonia [11,12,13]. Whereas chest CT scans carry a higher sensitivity in detecting lung involvement from the early phase of the disease [14,15], portable CXRs offer the undisputed advantage of minimizing the risk of cross-infection and reducing the movement of patients [13]; simultaneously, CXR scans demonstrate an overall balanced accuracy in diagnosing COVID-19 pneumonia in the acute care setting [16,17]. COVID-19 features on chest radiographs have been extensively described [18,19] and previous studies examined the predictive power of several CXR scores in COVID-19 pneumonia [12,20,21,22,23]. Most CXR scores included only qualitative information regarding the distribution and extension of pulmonary infiltrates [12,20,21,22]; in comparison, the recently validated ISARIC 4C Deterioration score [24] had the advantage of integrating both clinical and radiological data and was able to predict in-hospital clinical deterioration and death among hospitalized adults with COVID-19. However, radiological information was in this case limited to the presence of pulmonary infiltrates, with no other qualitative detail of their distribution and severity; moreover, currently none of the abovementioned scoring systems were specifically validated in the geriatric setting. In addition, assessment of frailty status was often not considered despite its recognized prognostic importance in hospitalized older adults with COVID-19 [8,9].

Although the frailty and severity of radiological involvement seems to be the expression of two different health status dimensions, described as a condition of increased vulnerability to poor resolution of homeostasis following stress and as a measure of the severity of lung involvement, their combined evaluation may help in capturing the overall risk of death in older patients with COVID-19.

For this reason, the aims of our study were to (a) evaluate the association between CXR score, frailty, clinical symptoms, and in-hospital mortality in a selected population of older hospitalized patients with COVID-19; (b) to compare predictive accuracy of in-hospital CXRs, frailty, and clinical symptoms in the same population; and to assess which of them might be better implemented in standard clinical practice to improve prognostic risk stratification.

## 2. Materials and Methods

### 2.1. Study Design and Inclusion Criteria

This was a retrospective observational study including 122 patients with a confirmed diagnosis of COVID-19 admitted to the acute geriatric ward of an Italian hospital from 1 March to 30 April 2020. Inclusion criteria were the following: patients aged over 65 years, SARS-CoV-2 infection (COVID-19) confirmed by reverse transcription polymerase chain reaction, and CXR performed immediately at the hospital admission. The study was approved by the Ethics Committee of INRCA IRCCS. Demographic, clinical, and laboratory data were extracted from electronic health records. Clinical data included symptoms and signs of infection such as fever, cough, dyspnea, diarrhea, nausea, and vomit. Frailty was graded according to the Rockwood Clinical Frailty Scale (CFS) that evaluates patient functional abilities 2 weeks before hospital admission and was specifically validated in the population of individuals of 65 years of age or more [25]. The CFS is an ordinal scale that ranks frailty from 1 to 9 (from being very fit to terminally ill), with higher scores indicating progressively higher degrees of frailty; patients with a CFS score > 4 were considered to be frail.

All patients underwent anteroposterior (AP) CXRs at hospital admission, performed directly in the isolation wards through portable X-ray units (GE VMX Mobile X-Ray). Two radiologists independently reviewed each admission of CXRs for the presence of consolidation, ground-glass opacities, reticular opacities, and pleural effusion according to the Fleischner Society glossary of terms [26]. Radiological involvement of lung parenchyma related to COVID-19 was described according to (a) the distribution of the disease (mostly peripheral or perihilar predominance); (b) the laterality of findings (unilateral or bilateral involvement); and (c) the predominance (upper, lower, or diffuse). In order to quantify the extension of pulmonary findings, a simplified version of the Radiographic Assessment of Lung Edema (RALE) severity score was used [18,27]. We chose this score as it has been proven to identify changes in the course of COVID-19, even though the radiologist assesses the lungs as a whole without dividing them into sectors. This allowed us to accelerate patients’ evaluation in conditions of high workflow burden. According to this adapted score, which was previously validated for COVID-19 infection [18], each lung was classified for the extension of involvement by consolidation, ground-glass opacities, and reticular opacities from 0 to 4 (0 = no involvement; 1 = <25%; 2 = 25–50%; 3 = 50–75%; and 4 = ≥75% of involvement), and the scores of both lungs were summed with a maximum value of 8 (an example can be seen in Figure 1).

### 2.2. Outcome

The outcome of the present study was in-hospital mortality. Patients who died were censored at the day of death, while survivors were censored at the day of discharge.

### 2.3. Statistical Analysis

Demographic, clinical, radiological, and laboratory characteristics of patients, both survivors and non-survivors, were compared by Student’s *t*-test or Mann–Whitney U test when appropriate for continuous variables and chi-square test for categorical ones. The association between each variable and mortality was explored by unadjusted Cox proportional hazard models. The CXR score, frailty, and variables significantly associated with the outcome in preliminary models were included in multivariable analysis. Five multivariable Cox proportional hazard models were built to obtain adjusted estimates of the association between exposure variables and the study outcome. The accuracy of exposure variables in predicting mortality was estimated by the Area Under the Receiver Operating Characteristic Curve (ROC). Finally, we investigated the additive effect of the CFS and other significant predictors on the predictive ability of the CXR score. Changes in Area Under the Curve (AUC) and categorical net reclassification index (NRI) with 1000 bootstrap samples to estimate 95% CIs were calculated. Statistical analysis was conducted using the Stata 15.1 Software Package for Windows (StataCorp, College Station, TX, USA).

## 3. Results

General characteristics of patients divided according to in-hospital mortality are reported in Table 1.

The study population consisted of 122 patients aged 87.1 ± 6.0 years with a slight female gender predominance (*n* = 67, 54.9%). Overall, 67 out of 122 patients (54.9%) died during hospital stay, with higher rates among women (53.7%). Patients who died were characterized by higher CXR and CFS scores, and there was a greater prevalence of dementia and congestive heart failure compared to the survivors (*p* < 0.05). Among the symptoms, fever and dyspnea at presentation were significantly more prevalent among patients who died.

Baseline chest radiography was positive in 84 patients with a CXR-sensitivity of 68.8%. Ground-glass opacities were the most common finding (65.5%), followed by reticular opacities (20.2%) and consolidation (14.3%). Peripheral distribution (57.1%) and lower zone distribution (69.0%) were the more common locations and most patients had bilateral involvement (56.0%). The CXR score significantly differed between survivors and non-survivors: while among survivors the maximum CXR score was four, patients who died had CXR scores ranging from zero to eight; moreover, all patients with a total CXR score greater than four at baseline chest radiography (*n* = 17) had fatal outcomes.

Unadjusted Cox regression analysis demonstrated that age, CXR score, CFS, congestive heart failure, dementia, fever, and dyspnea, and abnormal procalcitonin values were significantly associated with in-hospital mortality while stroke and comorbidity scores were nearly significantly associated with in-hospital mortality (Table 1). The above variables were included in the multivariable fully adjusted Cox proportional hazard models and four main predictors of in-hospital mortality were finally identified. In Model 1, including age, male gender, CXR score, CFS, and comorbidity score, the variables associated with mortality were found to be the CXR score (HR: 1.16; 95% CI 1.04–1.28), male gender (HR: 1.71; 95% CI 1.01–2.89), and CFS (HR: 1.27; 95% CI 1.09–1.47). Data were similar for Model 2, including single diagnoses instead of the comorbidity score. Conversely, in the models adjusted with the inclusion of either clinical or laboratory data, CFS and fever were the only significant predictors of the outcome (Table 2).

The Receiver Operating Characteristic curves for in-hospital mortality (Table 3) illustrated that the CXR score was a predictor of a fatal outcome in our study cohort of inpatients aged 70 to 101 years old with good accuracy (AUC = 0.70), slightly higher than that of the CFS (AUC = 0.67) and presence of fever (AUC = 0.61). Net reclassification analysis demonstrated that adding the CFS to the CXR score significantly improved the prediction of in-hospital mortality (continuous NRI = 0.355, 95% CI = 0.065–0.788; ΔAUC = 0.080, 95% CI = 0.006–0.153; *p* = 0.033). The addition of both CFS and presence of fever to the CXR score further improved the prediction of in-hospital mortality (continuous NRI=0.460, 95% CI = 0.102–0.888; ΔAUC = 0.117, 95% CI= 0.041–0.192; *p* = 0.003) in comparison to the model using only the CXR score.

The distribution of the CXR score in each CFS category by death and survival is displayed in Figure 2.

In patients who survived, the CXR score ranged from zero to four and was distributed in all CFS categories. Among patients who died, severely frail ones (CFS score 7–9) had a median CXR score of two, which was lower than that (4) of mildly or moderately frail patients (CFS score 1–6).

## 4. Discussion

Our study demonstrated that the CXR score, frailty status, and presence of fever were significant predictors of in-hospital mortality among older hospitalized patients with COVID-19. The strength of the association between either the CXR score or presence of fever and mortality was slightly reduced after introducing the CFS into the analysis. However, net reclassification analysis demonstrated that the model combining the CFS, CXR score, and presence of fever predicted the outcome with better accuracy compared to single risk factors. This may underline the importance of a multidimensional assessment including frailty, clinical, and radiological features when assessing hospitalized older patients with COVID-19.

This is the first study specifically comparing the predictive ability of frailty, radiological findings, and clinical data in hospitalized COVID-19 individuals aged 65 years or older. Older individuals represent a cluster of patients at higher risk for developing life-threatening respiratory failure related to COVID-19 due to the severity of lung involvement, immunosenescence and multimorbidity [28], and frailty. Frailty itself may contribute to increased vulnerability to more severe disease presentations.

As expected, frailty was a significant predictor of death in our study as well. In fact, patients with an increased CFS score were at a higher risk of death independent of the CXR findings. Compared to other frailty tools, the Rockwood CFS has the advantage of being specifically validated in older hospitalized people. Furthermore, it was suggested by the National Institute for Health and Care Excellence (NICE) guidelines for the assessment of older patients with COVID-19 [29] and proven to accurately predict in-hospital outcomes in this population [8,30]. Among clinical symptoms, fever was the only symptom to be significantly associated with the outcome in the study, maintaining its predictive weight in four out of five fully adjusted models, second only to the CFS. This result confirms previous evidence regarding the prognostic weight of fever and respiratory symptoms in hospitalized older patients with COVID-19 [10].

Radiological involvement of lung parenchyma due to COVID-19 pneumonia was demonstrated to be a marker of disease severity [21,31] and a predictor of poor outcomes in several hospitalized cohorts [21,22,23,31] but its prognostic weight in the geriatric population was not evaluated before. The features of radiological COVID-19 lung involvement in our cohort were similar to those reported in recent literature, including ground-glass opacities, peripheral distribution, lower zone distribution, and bilateral involvement [18,19]. Sensitivity of CXRs performed at hospital admission was about 68.8% in accordance with previous studies [12,18,22]. The CXR scores predicted in-hospital death with good accuracy (AUC: 0.70) and all patients with an overall score greater than four died during hospital stay. However, the association with mortality was decreased in models including the CFS, apart from those including the CXR score. This could be explained by the fact that CFS and CXR scores appeared to be independent from each other, capturing two different health status dimensions. In fact, the radiological severity of the disease did not increase with increasing frailty and severely frail patients died independently from the CXR score. NRI analysis finally illustrated that an integrated prognostic model combining the CFS, CXR score, and presence of fever in geriatric inpatients with COVID-19 yielded the highest prognostic accuracy in relation to in-hospital mortality (AUC: 0.80).

Our findings confirm the importance of both the radiological severity of COVID-19 pneumonia and frailty status in predicting poor outcomes in hospitalized older patients with COVID-19. It is arguable that, although these two factors were independent from each other, their combined evaluation may aid in improving prognostic risk stratification. From a clinical perspective, this relevant finding suggests the need of implementing multidimensional assessment integrating both clinical and radiological data in the acute geriatric setting; indeed, having easy-to-use diagnostic scores such as the CXR score and CFS may help accelerate the identification of more vulnerable older patients requiring targeted treatment approaches.

The limitations of this study are worth mentioning. Firstly, the retrospective study design and lack of a non-COVID-19 group may have limited the evaluation of sensitivity and specificity of the CXR score. Secondly, the small sample size may have decreased the precision of the estimates and did not allow for the estimation of a fully adjusted model in order to avoid overfitting issues; thus, other larger studies are necessary to validate these findings. Thirdly, in our study, we applied a visual evaluation of radiographic findings that may have influenced final results; in this regard, it would be desirable to implement artificial intelligence to increase the accuracy of CXR image analysis.

## 5. Conclusions

In conclusion, the CFS, CXR score, and presence of fever were the main predictors of in-hospital mortality in our cohort of hospitalized older patients with a confirmed diagnosis of COVID-19. The model integrating the three risk factors yielded the highest prognostic accuracy, which may be helpful for clinicians in identifying high-risk patients needing more intensive and tailored interventions.

## Figures and Tables

**Figure 1 jcm-10-02965-f001:**
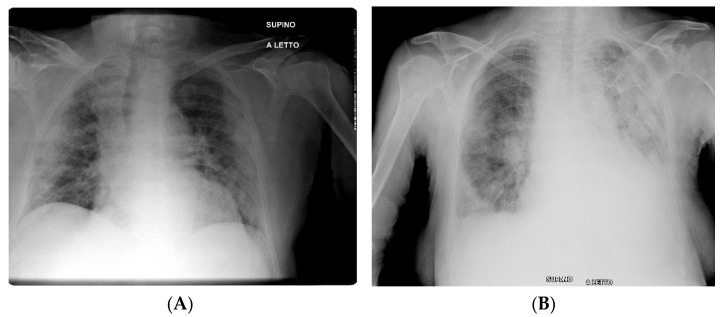
Examples of CXR scores in two patients with COVID-19 pneumonia. (**A**) presents consolidation with basal, peripheral, and bilateral predominance (right lung score + left lung score = total score; 3 + 3 = 6). (**B**) presents areas of consolidation and ground-glass opacity with subpleural and basal predominance in right lung, and diffuse areas of consolidation and ground-glass opacity in left lung (the calculation right lung score + left lung score = total score; 3 + 4 = 7).

**Figure 2 jcm-10-02965-f002:**
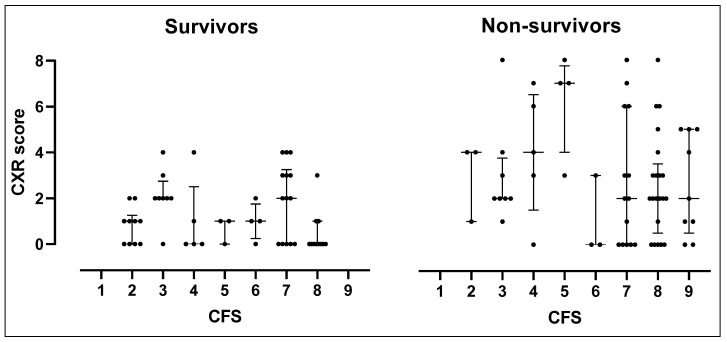
Column plot of the RX score in each CFS group and by death.

**Table 1 jcm-10-02965-t001:** Sample description.

	All *n* = 122	Survivors *n* = 55	Non-Survivors *n* = 67	*p*	Unadjusted HR (95% CI)
Male gender, *n*(%)	55 (45.1%)	24 (43.6%)	31 (46.3%)	0.771	1.03 (0.63–1.66)
Age, mean ± SD	87.1 ± 6.0	86.1 ± 6.7	87.9 ± 5.3	0.089	1.06 (1.01–1.10)
CXR score, median (IQR), range	2 (3), 0–8	1 (2), 0–4	2 (4), 0–8	<0.001	1.17 (1.06–1.30)
CFS, median (IQR), range	7 (4), 2–9	6 (4), 2–8	7 (3), 2–9	0.001	1.29 (1.13–1.46)
Diagnoses					
Hypertension, *n*(%)	95 (78.5%)	43 (78.2%)	52 (78.8%)	0.936	1.10 (0.61–1.99)
Diabetes, *n*(%)	32 (26.2%)	11 (20.0%)	21 (31.3%)	0.156	1.42 (0.85–2.39)
Stroke, *n*(%)	17 (13.9%)	5 (9.1%)	12 (17.9%)	0.162	1.86 (0.99–3.50)
Cancer, *n*(%)	13 (10.7%)	4 (7.3%)	9 (13.4%)	0.273	1.36 (0.67–2.75)
COPD, *n*(%)	26 (21.3%)	13 (23.6%)	13 (19.4%)	0.570	0.83 (0.45–1.52)
Asthma, *n*(%)	4 (3.3%)	3 (5.5%)	1 (1.5%)	0.227	0.42 (0.06–3.06)
Angina, *n*(%)	7 (5.8%)	3 (5.6%)	4 (6.0%)	0.923	0.86 (0.31–2.38)
Myocardial Infarction, *n*(%)	21 (17.6%)	9 (17.0%)	12 (18.2%)	0.864	1.00 (0.53–1.88)
Atrial Fibrillation, *n*(%)	30 (24.8%)	10 (18.2%)	20 (30.3%)	0.124	1.31 (0.77–2.23)
CHF, *n*(%)	25 (20.8%)	6 (11.1%)	19 (28.8%)	0.018	1.76 (1.02–3.05)
Dementia, *n*(%)	69 (56.6%)	24 (43.6%)	45 (67.2%)	0.009	2.31 (1.38–3.86)
CKD, *n*(%)	42 (35.0%)	17 (31.5%)	25 (37.9%)	0.465	1.22 (0.74–2.01)
Previous ADRs, *n*(%)	9 (7.4%)	4 (7.3%)	5 (7.5%)	0.968	0.85 (0.34–2.14)
Concomitant Bacterial Infections, *n*(%)	10 (8.4%)	5 (9.3%)	5 (7.7%)	0.759	0.87 (0.35–2.17)
Comorbidity score, median(IQR), range	5 (1), 3–12	5 (2), 4–11	5 (1), 3–12	0.085	1.14 (0.99–1.32)
Symptoms					
Fever, *n*(%)	64 (52.5%)	22 (40.0%)	42 (62.7%)	0.013	1.70 (1.03–2.81)
Cough, *n*(%)	39 (33.1%)	19 (34.5%)	20 (31.7%)	0.747	0.81 (0.48–1.39)
Dyspnea, *n*(%)	88 (72.1%)	29 (52.7%)	59 (88.1%)	<0.001	3.38 (1.61–7.09)
Diarrhea, *n*(%)	11 (9.0%)	6 (10.9%)	5 (7.5%)	0.508	0.61 (0.24 1.54)
Nausea, *n*(%)	5 (4.1%)	4 (7.3%)	1 (1.5%)	0.109	0.34 (0.05–2.48)
Vomit, *n*(%)	4 (3.3%)	2 (3.6%)	2 (3.0%)	0.841	0.48 (0.11–2.04)
Abnormal lab parameters					
WBC (×10^3^/µL), *n*(%)	51 (41.8%)	21 (38.2%)	30 (44.8%)	0.462	1.24 (0.76–2.02)
Lymphocytes (×10^3^/µL), *n*(%)	67 (54.9%)	32 (58.2%)	35 (52.2%)	0.512	0.70 (0.42–1.18)
CPK (U/L), *n*(%)	44 (36.4%)	25 (46.3%)	19 (28.4%)	0.041	1.89 (0.74–4.82)
LDH (U/L), *n*(%)	76 (62.8%)	27 (50.0%)	49 (73.1%)	0.009	2.55 (0.98–6.65)
eGFR (mL/min/1.73 m^2^), *n*(%)	105 (86.1%)	44 (80.0%)	61 (91.0%)	0.080	2.26 (0.97–5.26)
NLR, *n*(%)	77 (63.1%)	31 (56.4%)	46 (68.7%)	0.161	1.33 (0.79–2.23)
CRP (mg/dL), *n*(%)	115 (94.2%)	49 (89.1%)	66 (98.5%)	0.080	0.52 (0.07–3.80)
D-dimer (ng/mL), *n*(%)	116 (95.9%)	53 (98.2%)	63 (94.0%)	0.258	1.09 (0.39–3.05)
Procalcitonin (ng/mL), *n*(%)	73 (59.8%)	28 (50.9%)	45 (67.2%)	0.125	2.61 (1.17 5.83)
IL-6 (pg/mL), *n*(%)	98 (80.3%)	38 (69.1%)	60 (89.5%)	0.005	1.48 (0.58–3.75)

*Abbreviations*: COPD = Chronic Obstructive Pulmonary Disease; CHF = Congestive Heart Failure; CKD = Chronic Kidney Disease; ADR = Adverse Drug Reaction; CFS = Clinical Frailty Scale; WBC = White Blood Cell; CPK = Creatine Phosphokinase; LDH = Lactate Dehydrogenase; eGFR = estimated Glomerular Filtration Rate; NLR = Neutrophils Lymphocytes Ratio; CPR = C Reactive Protein; IL-6 = Interleukin 6; HR (95% CI) = Hazard Ratio (95% Confidence Interval); SD = standard deviation; and IQR = InterQuartile Range. *Notes*: *p*-value from χ^2^ test, *t*-test, or Mann–Whitney U test as appropriate. *Normal values for lab parameters*: 1 × 10^3^/µL ≤ WBC ≤ 4 × 10^3^/µL; 1 × 10^3^/µL ≤ Lymphocytes ≤ 4 × 10^3^/µL; 26 U/L ≤ CPK≤ 192 U/L (female), 39 U/L ≤ CPK ≤ 308 U/L (male); LDH ≤ 280 U/L; eGFR < 60mL/min/1.73 m^2^; 1 ≤ NRL ≤ 3.53; CRP ≤ 1mg/dL; D-dimer ≤ 250 ng/mL; Procalcitonin ≤ 0.15 ng/mL; and IL-6 ≤ 15 pg/mL.

**Table 2 jcm-10-02965-t002:** Cox proportional hazards models for CXR, CFS, clinical, and laboratory data for death during hospitalization.

*n* = 122	Model 1	Model 2	Model 3	Model 4	Model 5
	HR (95% CI)	HR (95% CI)	HR (95% CI)	HR (95% CI)	HR (95% CI)
CXR score	1.16 (1.04–1.28)	1.14 (1.01–1.27)	1.12 (0.99–1.26)	1.11 (0.99–1.26)	1.10 (0.97–1.25)
Age	1.04 (0.99–1.10)	1.04 (0.99–1.10)	1.04 (0.98–1.10)	1.04 (0.98–1.10)	1.04 (0.98–1.11)
Male gender	1.71 (1.01–2.89)	1.70 (1.01–2.87)	1.71 (0.99–2.94)	1.73 (1.01–2.95)	1.87 (1.05–3.32)
CFS	1.27 (1.09–1.47)	1.21 (1.01–1.45)	1.28 (1.09–1.49)	1.24 (1.03–1.49)	1.23 (1.04–1.46)
Comorbidity score	1.12 (0.96–1.31)	-	1.12 (0.95–1.32)	-	1.11 (0.94–1.32)
Stroke	-	1.38 (0.69–2.75)	-	1.15 (0.57–2.31)	-
CHF	-	1.19 (0.67–2.11)	-	1.17 (0.66–2.08)	-
Dementia	-	1.29 (0.64–2.58)	-	1.11 (0.55–2.21)	-
Fever	-	-	1.75 (1.03–2.97)	1.71 (1.00–2.93)	-
Dyspnea	-	-	1.59 (0.70–3.64)	1.60 (0.69–3.66)	-
Abnormal CPK (U/L)	-	-	-	-	0.99 (0.39–2.54)
Abnormal LDH (U/L)	-	-	-	-	2.29 (0.82–6.38)
Abnormal D-dimer (ng/mL)	-	-	-	-	0.69 (0.21–2.23)
Abnormal Procalcitonin (ng/mL)	-	-	-	-	1.72 (0.77–3.84)
Abnormal IL-6 (pg/mL)	-	-	-	-	1.27 (0.42–3.86)

*Abbreviations*: CFS = Clinical Frailty Scale; CHF = Congestive Heart Failure; CI = Confidence Interval; CPK = Creatine phosphokinase; CXR = Chest X-ray; HR = Hazard Ratio; IL-6 = Interleukin 6; and LDH = Lactated Dehydrogenase. *Note*: Model 1: including CXR score, age, male gender, CFS, and comorbidity score. Model 2: alike to Model 1 but using stroke, CHF, and dementia instead of comorbidity score. Model 3: Model 1 + fever and dyspnea. Model 4: Model 2 + fever and dyspnea. Model 5: Model 1 + abnormal CPK, abnormal LDH, abnormal D-dimer, and abnormal IL-6.

**Table 3 jcm-10-02965-t003:** Accuracy of the CXR score and net reclassification analysis for death during hospitalization.

Outcome	Addition	AUC (95% CI)	Overall NRI (95% CI)	ΔAUC (95% CI)	*p*
Death (*n* = 122)		0.701 (0.611–0.790)			
	CFS		0.355 (0.065–0.788)	0.080 (0.006–0.153)	0.033
	Fever		0.454 (−0.336–0.794)	0.026 (−0.350–0.086)	0.410
	CFS and Fever		0.460 (0.102–0.888)	0.117 (0.041–0.192)	0.003

*Abbreviations*: AUC = Area Under the Curve; CFS = Clinical Frailty Scale; CI = Confidence Interval; CXR = Chest X-ray; and NRI = Net Reclassification Improvement.

## Data Availability

The data presented in this study are available upon request from the corresponding author. The data are not publicly available as they contain information that could compromise the privacy of the research participants.

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
