# Peer review of "Chest X-ray Score and Frailty as Predictors of In-Hospital Mortality in Older Adults with COVID-19"

_jcm, 2021, doi:10.3390/jcm10132965_

Round 1

Reviewer 1 Report

Thank you for sending your work for consideration in JCM. This piece is highly pertinent at the moment and should be in the minds of the readership.
There are a few points that ought to be considered to improve what is fundamentally a good article. If you can address these, the work will be highly useful.
The ideas are given below:

  • expand the bibliography a little by introducing the following articles:

-Borakati, A.; Perera, A.; Johnson, J.; Sood, T. Diagnostic accuracy of X-ray versus CT in COVID-19: A propensity-matched database study. BMJ Open 2020, 10, e042946. [CrossRef]

-Brogna B, Bignardi E, Brogna C, Volpe M, Lombardi G, Rosa A, Gagliardi G, Capasso PFM, Gravino E, Maio F, Pane F, Picariello V, Buono M, Colucci L, Musto LA. A Pictorial Review of the Role of Imaging in the Detection, Management, Histopathological Correlations, and Complications of COVID-19 Pneumonia. Diagnostics (Basel). 2021 Mar 4;11(3):437. doi: 10.3390/diagnostics11030437. PMID: 33806423; PMCID: PMC8000129.

-Maroldi, R.; Rondi, P.; Agazzi, G.M.; Ravanelli, M.; Borghesi, A.; Farina, D. Which role for chest x-ray score in predicting the outcome in COVID-19 pneumonia? Eur. Radiol. 2020, 2, 1–7. [CrossRef]

-Cozzi, A.; Schiaffino, S.; Arpaia, F.; Della Pepa, G.; Tritella, S.; Bertolotti, P.; Menicagli, L.; Monaco, C.G.; Carbonaro, L.A.; Spairani, R.; et al. Chest x-ray in the COVID-19 pandemic: Radiologists’ real-world reader performance. Eur. J. Radiol. 2020, 132, 109272. [CrossRef]

-Brogna, B.; Bignardi, E.; Brogna, C.; Alberigo, M.; Grappone, M.; Megliola, A.; Salvatore, P.; Fontanella, G.; Mazza, E.; Musto, L. Typical CT findings of COVID-19 pneumonia in patients presenting with repetitive negative RT-PCR. Radiography 2020. [CrossRef]

-Warren, M.A.; Zhao, Z.; Koyama, T.; Bastarache, J.A.; Shaver, C.M.; Semler, M.W.; Rice, T.W.; Matthay, M.A.; Calfee, C.S.; Ware, L.B. Severity scoring of lung oedema on the chest radiograph is associated with clinical outcomes in ARDS. Thorax 2018, 73, 840–846. [CrossRef]

-Schiaffino, S.; Tritella, S.; Cozzi, A.; Carriero, S.; Blandi, L.; Ferraris, L.; Sardanelli, F. Diagnostic Performance of Chest X-Ray for COVID-19 Pneumonia During the SARS-CoV-2 Pandemic in Lombardy, Italy. J. Thorac. Imaging 2020, 35, W105–W106. [CrossRef] [PubMed

      2) briefly clarify why the RALE Score was used and not other CXRs Scores

       3) stress the limits of the Chest X-ray , even briefly and the reasons that led the authors to use Chest x-ray and not Chest CT

      4) Introduce some x-ray images, illustrating how the CXR Score is calculated

Reviewer 2 Report

Thank you for the opportunity to review the manuscript titled “Chest X-ray Score and Frailty as predictors of in-hospital mortality in older adults with COVID-19”. This cross-sectional study aimed to evaluate the association between potential predictive factors and COVID-19-related in-hospital mortality. The manuscript is generally well written and clearly presented. I only offer a few minor suggestions to further improve readability of the manuscript (see below).

  1. Line 27 & Line 29: Please ensure that abbreviations (i.e. RT-PCR and CFS) are defined on first use in the abstract.
  2. Line 83 & Line 93 & Line 96 & Line 102 & Line 120 & Line 250: To improve readability, I suggest removing abbreviations (i.e. RT-PCR, CO, GGO, RO, RALE, AUROC, AI) that are used infrequently, especially given that this manuscript features a relatively large number of abbreviations
  3. Line 115 & Line 121: The letter ‘C’ is missing in the abbreviation ‘CRX’.
  4. Lines 117-118: It is not made clear why five separate multivariable models were built. I think it will suffice to simply present the final fitted model following the forward selection process (presumably the current Model 1). If there is a particular interest in evaluating the change in predictive accuracy from adding a particular variable to the final fitted model, then that needs a clear justification.
  5. Line 128: For Table 1, please ensure the column heading for non-survivors is consistent with the terminology used in the manuscript text. That is the terms ‘dead’, ‘deceased’, and ‘non-survivor’ are used interchangeably at different time. To improve readability, select one term and use it consistently throughout the manuscript.
  6. Line 128: Why present hazard ratios (HS) adjusted for age and gender in Table 1? I suggest presenting univariable unadjusted HRs and leave the adjustments for the subsequent multivariable model fitting. In any event, it does not make sense to present a HR ratio for gender that is adjusted for gender, nor a HR for age that is adjusted for age.
  7. Line 128: The p-value for CXR score cannot be 0.000, please use inequality symbols for very low p-vales instead (e.g. p < 0.001).
  8. Lines 139-151: Please ensure that all proportions are reported consistently in terms of number of decimal places and with or without space before the percentage symbol.
